# Influence of Smoking and Paprika Spice on the Content of Polycyclic Aromatic Hydrocarbons (PAHs) in the Traditional Spanish Smoked Sausage ‘Botillo del Bierzo’

**DOI:** 10.3390/foods13244089

**Published:** 2024-12-17

**Authors:** Jaime R. Fuertes-Martínez, Marcos Guerra, Álvaro Rodríguez-González, Héctor del Valle-Herrero, José B. Valenciano, Víctor Marcelo

**Affiliations:** 1Departamento Ingeniería y Ciencias Agrarias, Universidad de León, Avenida de Astorga s/n, 24401 Ponferrada, Spain; jr.fuertes@unileon.es; 2Grupo Universitario de Investigación en Ingeniería y Agricultura Sostenible (GUIIAS), Escuela de Ingeniería Agraria y Forestal, Campus de Ponferrada, Universidad de León, Avenida de Astorga s/n, 24401 Ponferrada, Spain; mgues@unileon.es; 3Grupo Universitario de Investigación en Ingeniería y Agricultura Sostenible (GUIIAS), Instituto de Medio Ambiente Recursos Naturales y Biodiversidad (INMARENBIO), Escuela de Ingeniería Agraria y Forestal (EIAF), Universidad de León, Avenida de Portugal 41, 24071 León, Spain; alrog@unileon.es; 4Laboratorio Físico-Químico, Instituto Tecnológico Agrario de Castilla y León, Consejería de Agricultura y Ganadería, Ctra. Burgos Km 119, Finca Zamadueñas, 47071 Valladolid, Spain; ita-valherhe@itacyl.es; 5Departamento Ingeniería y Ciencias Agrarias, Universidad de León, Avenida de Portugal 41, 24071 León, Spain; joseb.valenciano@unileon.es

**Keywords:** botillo, Botillo del Bierzo, fermented sausages, polycyclic aromatic hydrocarbons, smoked meat products, smoked paprika

## Abstract

The content of polycyclic aromatic hydrocarbons (PAHs) in a traditional Spanish smoked sausage known as ‘Botillo del Bierzo’ was analysed. The determination and quantification of PAH4 (the sum of benzo[a]pyrene, benz[a]anthracene, benzo[b]-fluoranthene and chrysene) in the smoked sausage were performed using GC–MS. The results showed that smoking ‘Botillo del Bierzo’ for 24 h contributes very little to the increase in PAH levels in the product, with no significant differences (*p* < 0.05). The paprika used in the production of botillo also contributes to PAH content, depending on the paprika’s production process (smoked, low-smoked, mixed paprika (smoked + unsmoked or low-smoked + unsmoked), and unsmoked), with significant differences among them (*p* < 0.05). Using paprika with a lower PAH content (unsmoked or mixed) enables the production of ‘Botillo del Bierzo’, in accordance with Protected Geographical Indication specifications, with a lower PAH content while maintaining the organoleptic characteristics provided by the paprika. The contamination levels detected in ‘Botillo del Bierzo’ do not pose a risk to consumer health, as the benzo[a]pyrene and PAH4 values are well below the regulatory limits of 5 μg/kg and 30 μg/kg, respectively.

## 1. Introduction

The smoking of meat and meat products has been used for thousands of years as a method of food preservation. Currently, 40–60% of meat products are smoked; smoking is mainly used for the effects of various compounds present in the smoke, which enhance the flavour of meat products, giving them a distinctive aroma, taste, and colour that is widely demanded by consumers [1,2,3,4].

‘Botillo’ is a traditional Spanish raw-cured sausage primarily produced in the regions of Castilla y León, Galicia, and Asturias (Northwest Spain). It is a product highly sought after by consumers, especially in the Bierzo area, which has led to its evolution from artisanal, homemade production to industrial manufacturing under the protection of a European quality label (Protected Geographical Indication (PGI) of ‘Botillo del Bierzo’) since 2000 [5,6]. This PGI has a specification document that regulates its production [7]. ‘Botillo del Bierzo’ is made from pieces of ribs, tails and other pork bones with their fleshy parts. The meat is mixed with salt, sweet and spicy paprika, garlic, and other spices then left to stand before being stuffed into pork caecum. It then undergoes a heating–smoking process for 1 day, followed by drying and ripening for at least 2 days. It is then ready for consumption, which takes place after cooking [7].

The smoking process is carried out through the combustion of *Quercus robur* L. or *Quercus ilex* L. wood. The incomplete combustion of wood during the smoking process can produce a considerable amount of polycyclic aromatic hydrocarbons (PAHs), with their content varying depending on the method and conditions of smoking [4,8,9,10]. PAHs are organic compounds that pose a potential risk to human health, as they can cause DNA damage, mutations, and cancer in the human body [2,11,12,13]. For these reasons, in the interest of public health, the European Union has enacted regulations based on the recommendations of the European Scientific Committee on Food (SCF) [14]. Initially, EC Regulation No. 208/2005 [15] was introduced, followed in 2006 by Commission Regulation No. 1881/2006, which set a maximum limit for benzo[a]pyrene (BaP) content at <5 μg/kg [16]. Since 2008, the European Food Safety Authority (EFSA) has recommended that four PAHs (benzo[a]pyrene, benz[a]anthracene, benz[b]fluoranthene, and chrysene) be considered more suitable indicators of PAH presence in the diet than BaP alone [17,18]. Subsequent amendments reduced the maximum allowable limit for BaP to <2 μg/kg and for PAH4 (benz[a]anthracene, chrysene, benz[b]fluoranthene and benzo[a]pyrene) in smoked meat products to <12 μg/kg [19]. This reduction led to protests from producers of traditionally smoked meat and fish products, as smoking practices in these cases cannot be modified without significantly altering the organoleptic characteristics of the food. However, Commission Regulation No. 1327/2014 [20] temporarily increased the limit for BaP to <5 μg/kg and for PAH4 to <30 μg/kg for products intended for national consumption. This temporary exception was made permanent without a time limit by Commission Regulation No. 2020/1255 [21]. Finally, Regulation 915/2023 [22] repealed Regulation 1881/2006 but maintained the BaP limit of <2.0 μg/kg and the PAH4 limit of <12.0 μg/kg for products marketed within the EU, while keeping the BaP limit of <5.0 μg/kg and PAH4 limit of <30.0 μg/kg for traditionally smoked meat products consumed within the producing member state.

On the other hand, it is important to note that paprika is one of the spices used in the production of ‘Botillo del Bierzo’. Paprika is a product obtained from dehydrated and milled fruits of certain varieties of red peppers (*Capsicum annuum* L.) [23]. There are different drying methods to produce this spice. In Spain, two main regions are known for paprika production: La Vera (Extremadura) and La Huerta (Murcia). In La Vera, the peppers are smoked and dried using oak or holm oak wood fires, while in Murcia, among other places, the peppers are sundried [24,25]. The smoking process provides organoleptic characteristics appreciated by consumers, but it can also result in the presence of undesirable substances such as PAHs in paprika [26,27]. In addition, smoked *Capsicum* fruits are exempt from the restrictions required for other smoked products, as the consumption of these spices is low [18,28].

The objective of this work was to demonstrate the impact of smoking on the PAH4 content in ‘Botillo del Bierzo’ as well as the influence of the paprika spice used during its production on the PAH4 content of ‘Botillo del Bierzo’.

## 2. Materials and Methods

### 2.1. Botillo del Bierzo Processing and Sampling

‘Botillo del Bierzo’, a traditional dry-cured sausage from Northwest Spain, was produced in various local factories using meat from commercial pig breeds. It is made from cut pieces of ribs (65–90%), tail pieces (10–20%), and vertebrae, along with other pork bones with their fleshy parts (tongue and cheek), comprising a maximum of 20%. The meat is mixed with salt (1.8–2.5% *w*/*w*), paprika (*Capsicum annuum* L.) paste (2.1–3.0% *w*/*w*), garlic (*Allium sativum* L.) paste (0.2–0.5% *w*/*w*), and other spices. The meat batter was stored under controlled conditions at 8–10 °C with 90% relative humidity, left to stand for 12–24 h, and then stuffed into pork caecum.

A total of 124 samples were taken from all PGI-certified ‘Botillo del Bierzo’ producers over four years (2019–2022), with both smoked and unsmoked samples analysed to determine their PAH4 content. Two experimental groups were established on the basis of smoking: (Uns) unsmoked and (S) smoked. Heating–smoking was generated by burning oak (*Quercus robur* L.) or holm oak (*Quercus ilex* L.) wood. Smoked ‘Botillo del Bierzo’ samples were directly exposed to smoke for one day (Figure 1), then left to dry and ripen for at least 2 days in a curing chamber under controlled conditions at 10–12 °C and 75–85% relative humidity. In all cases, once collected, the samples were transported to the laboratory under refrigeration (4 °C).

The period analysed for PAH4 in ‘Botillo del Bierzo’ was 2019–2022. Once the minimal influence of smoking on PAH levels in ‘Botillo del Bierzo’ was observed, an analysis of the paprika used in its production was conducted during the period of 2021–2024, along with further analysis of the produced botillos. During this period, 96 samples of paprika used as a spice in the production of botillos were analysed to assess its influence on the PAH4 content of the final product. Additionally, 104 samples of botillos made with different types of paprika were analysed. Four experimental groups were established on the basis of the type of paprika used in the production of ‘Botillo del Bierzo’: (S) smoked, (LS) low-smoked (50% shorter exposure time to smoke), (M) mixed paprika (smoked + unsmoked or low-smoked + unsmoked), and (Uns) unsmoked.

In order to prepare the samples for analysis, the outer casing was removed and discarded, and the bone was eliminated from the ‘Botillo’ pieces. The edible part was then thoroughly cut and ground using a mincer machine until a homogeneous mass was obtained.

### 2.2. Analytical Methods

The analytical determinations were carried out in quadruplicate. Moisture, dry matter (DM), fat, and protein (Kjeldahl Nx6.25) were determined using the ISO recommended procedures: ISO 1442:2023 [29], ISO 1444:1996 [30], and ISO 937:2023 [31], respectively.

### 2.3. PAH Determination

#### 2.3.1. Chemicals

Acetonitrile (≥99.9%, HPLC grade), ethyl acetate (≥99.8%, HPLC grade), and N-hexane (≥99.0%, HPLC grade) were acquired from DISMED S.A. (Gijón, Spain). Water from a water purification system (18 MΩ cm resistivity) was used in all experiments. Helium (BIP grade) was purchased from Carburos Metálicos S.A. (Cornellá de Llobregat, Spain). Anhydrous magnesium sulfate (≥97.0%, technical grade) and anhydrous sodium acetate (≥99.0%, PRS-CODEX grade) were purchased from Panreac (Barcelona, Spain). QuEChERS (Quick, Easy, Cheap, Effective, Rugged, and Safe) kits for PAH purification by dispersive solid-phase extraction (dSPE) were supplied by Aplicaciones Cromatográficas S.L. (Murcia, Spain) and consisted of 15 mL conical tubes with 500 mg of a zirconium-based phase. For gas chromatography–mass spectrometry (GC–MS) method development and validation, commercial standards (PAH Analyzer Calibration Kit vial 1 (Agilent Technologies, Inc., Santa Clara, CA, USA)) and deuterated commercial standards (PAH Analyzer Calibration Sample Kit Vial 2 (Agilent Technologies, Inc. Santa Clara, CA, USA) were used, both purchased from Aplicaciones Cromatográficas S.L. (Murcia, Spain). Among the deuterated standards, only chrysene-d12 was used as the internal standard, as it corresponds to a regulated PAH. Other deuterated compounds in the kit (e.g., naphthalene-d8, perylene-d12, and phenanthrene-d10) were not included as internal standards because their non-deuterated equivalents are not subject to current legislation. A sample from an interlaboratory (BIPEA/12-2444), containing a mixture of PAHs including group 4 PAHs with regulated values, was used for quality control.

#### 2.3.2. Preparation of Samples for GC–MS Analysis

For the extraction of the target analytes (benzo[a]pyrene, benz[a]anthracene, benzo[b]fluoranthene, and chrysene) from meat product and paprika samples, a solid-liquid extraction procedure was performed, following the steps listed below. (i) Approximately 0.5 g of anhydrous sodium acetate, 2 g of MgSO_4_, 2.50 ± 0.10 g of minced meat product samples, and 1.00 ± 0.10 g of paprika were weighed (Sartorius-Te124S analytical balance) and transferred into 50 mL polypropylene centrifuge tubes. Each sample was prepared in duplicate. (ii) A total of 300 µL of chrysene d12, used as the internal standard (ISTD) (0.2 µg/mL concentration), was added to obtain 0.150 µg/mL in each sample replicate. (iii) Tubes were labelled with the laboratory codes ‘+DOP’ (for the doped sample) and ‘SDOP’ (for the undoped sample). The weight of each duplicate was recorded on the primary datasheet. (iv) The sample labelled ‘+DOP’ was doped with 150 µL of the P5PAHs S.P.I. standard solution and kept capped for 30 min. (v) A total of 20 mL ethyl acetate/acetonitrile (20:80, *v*/*v*) was added to all tubes using a 25 mL dispenser. (vi) Samples were intensely agitated using a multivortex (Heidolph Multi Reax) for 10 min at room temperature, followed by centrifugation at 4000 rpm for 10 min at 4 °C in a Selecta Meditronic BL-S centrifuge (Barcelona, Spain).

Matrix cleanup was performed using QuEChERS dispersive solid-phase extraction. First, 5 mL of the extract was pipetted and transferred into a 15 mL commercial QuEChERS dZsep tube. The samples were then vigorously agitated using a multivortex (Heidolph Multi Reax, Schwabach, Germany) for 5 min at room temperature, followed by centrifugation at 4000 rpm for 10 min at 4 °C in a Selecta Meditronic BL-S centrifuge (Barcelona, Spain).

#### 2.3.3. Quantification of PAH by GC–MS

The chromatographic conditions of PAH analysis were based on Agilent application notes [32,33]. An Agilent 7890B gas chromatograph (Agilent Technologies Inc. Lexington, MA, USA) with a 5977B MSD (Agilent Technologies, USA) detector was used, along with a single quadrupole mass spectrometer (Agilent Technologies, USA) equipped with selective ion monitoring (SIM) mode for enhanced sensitivity. The supernatants collected from the cleanup process were transferred into tubes of the TurboVap LV equipment (Biotage, AB, Uppsala, Sweden) and evaporated to dryness under a gentle stream of N_2_ at 50 °C. The samples were then redissolved in 100 µL of ethyl acetate:hexane (1:1, *v*/*v*) and encapsulated in 1.5 mL vials with corresponding 200 µL inserts. Sample injections (2 µL) were performed in splitless mode, with a splitless time of 1.2 min, using a Splitless injector temperature of 320 °C (593.15 °K). A DB-EUPAH 122-9632 column (Agilent J&W) with dimensions of 30 m length, 0.25 mm internal diameter, and 0.25 μm film thickness was employed. Helium at a flow rate of 0.8 mL/min served as the mobile phase. The MS (5977B) ion source temperature was set to 325 °C (598.15 °K), and the transfer line was set at 340 °C (613.15 °K). The temperature program was as follows: initial hold for 1 min at 70 °C (343 °K); 25 °C/min to 195 °C (468.15 °K); 7 °C/min to 315 °C (588.15 °K).

Identification of the compounds was achieved through SCAN mode, scanning between two given masses and comparing the resulting mass spectra with the NIST library. Retention times were compared with data from Agilent applications [32,33]. Key parameters for identification included the presence and relative abundance of qualification ions in comparison with quantification ions.

After method optimisation, the retention times and qualification and quantification ions for the compounds benz[a]anthracene, chrysene d12 (ISTD), chrysene, benzo[b]fluoranthene, and benzo[a]pyrene were flagged. The retention times (in minutes) for each compound were 17.631, 17.781, 17.872, 21.244, and 22.952, respectively. The quantification ions (*m*/*z*) for each compound were 228.1, 240.1, 228.1, 252.1, and 252.1, respectively. Finally, the qualification ions (*m*/*z*) for each compound were 229.1, 241.1, 229.1, and 250.1, respectively.

For the quantification of the test compounds, a concentration-weighted least squares line (1/x) was developed between the ratio of the area of each compound to the ISTD area and the ratio of the theoretical concentration to the ISTD concentration. The concentration-weighted line (1/x) was used because a suitable calibration interval was needed to interpolate areas of high and low concentrations while minimising deviations in the recovery values of each calibration point.

For each batch of samples, full calibration was performed, and internal acceptance criteria were verified. These criteria ensured that the internal acceptance criteria were met (recovery values between 93 and 107% of each calibration point and a coefficient of determination, r2, greater than 0.992).

The ratio of the area of each sample in each analyte and the ISTD was interpolated on the line. The software provided the concentration in ng/mL, taking into account a dilution factor (aliquot collected in the extraction for QuEChERS use and the initial sample weight). The calculation of the dilution factor was performed using Equation (1):(1)$$\frac{\mu g}{kg}=\frac{ng}{mL}\times \frac{0.1(redissolution\;volume)\times 4(aliquot\;in\;dZsep)}{initial\;weight\;g}$$

Each sample, including one doped with a known concentration of the target compounds, was extracted in duplicate. During the quantification process, the recovery of these doped samples and the repeatability of the method were evaluated against the initial conditions established during the method validation. These criteria were used to ensure the acceptance of the analytical results.

For all calculations, Agilent Masshunter Quantitative Analysis (Agilent Technologies Inc. Lexington, MA, USA) for GC–MS Version B.08.00 software was used. The software facilitated the inclusion of the dilution factor, the ISTD calibration, and the recovery data for each calibration point.

#### 2.3.4. GC–MS Method Validation

To establish the quality of the proposed method for PAH determination in the samples under study by GC–MS, the following analytical characteristics [linearity, sensitivity, repeatability, reproducibility, recoveries, limits of detection (LOD), and quantification (LOQ)] were determined (Table 1).

For method validation, doped samples from smoked meat products with low PAH concentrations, along with one sample from a Bipea ring trial, were taken over three different days. Each day, five concentration levels of PAHs were analysed in triplicate. To create the different concentration levels, a dilution of ‘D.M. Calibration’ at 12 ng/mL for each PAH (0.12 mL of D.M. Calibration into 10 mL ethyl acetate/hexane 1:1, *v*/*v*), called ‘D.M. Validation’, was prepared (Figure 2).

For each sample, except for the Level 2 sample, doping was carried out using the specified volume [Volume D.M. Validation 12 (ng/mL) and Volume D.M. Internal 1] and concentration levels [individual PAH (ng/mL), I.P. (ng/mL), individual PAH (µg/kg), and sum PAH (µg/kg)], as described below for each level. Level 0 sample: 0.064, 250, 1.92, 50, 0.31, and 1.23, respectively. Level 1 sample: 0.2, 250, 6, 50, 0.96, and 3.84, respectively. Level 3 sample: 1, 250, 30, 50, 4.8, and 19.2, respectively. Level 4 sample: 2, 250, 60, 50, 9.6, and 38.4, respectively. Level 5 sample: 3, 250, 90, 50, 14.4, and 57.6, respectively.

The Level 2 sample (BIPEA/12-2444) was designated as the Bipea interlaboratory sample, and its certified values (result ± uncertainty) were as follows: benz[a]anthracene 3.0 ± 0.2 µg/kg, chrysene 4.1 ± 0.2 µg/kg, benzo[b]fluoranthene 3.1 ± 0.2 µg/kg, benzo[a]pyrene 3.3 ± 0.2 µg/kg. The total sum was 13.5 ± 0.7 µg/kg.

According to EC Regulation No. 836/2011 [34], the sampling and analysis methods for the official control of benzo[a]pyrene in food products established the minimum criteria accepted for the determination of PAH. The acceptance criteria required for the determination of PAH are described below. RSDr repeatability: [(observed RSDR)/(Horwitz estimated RSDR × 0.66) < 2]; reproducibility RSDR [(observed RSDR)/(Horwitz estimated RSDR) < 2]; recovery (50–120%); LOD ≤ 0.3 μg/kg in each substance; and LOQ ≤ 0.9 μg/kg in each substance.

For the calculation of RSDR, the Horwitz ratios (HORRATR) were calculated for each EC-regulated PAH by dividing the relative standard deviations of the mean concentrations for each analyte in the same spiked samples by the predicted value of 22%, as indicated by the modified Horwitz equation (for concentrations below 1.2 × 10^−7^) for trace analysis [35]. In order, they were found to be 0.440 (RSDR) and 0.290 (RSDr). These values are well below the upper limit of 2.000 [34].

### 2.4. Statistical Methods

A one-way ANOVA and a two-way ANOVA were performed using IBM SPSS Statistics version 29.0.2.0. The post hoc Tukey test was employed to compare the means (*p* < 0.05).

## 3. Results and Discussion

The analysis of the samples indicates that the ‘Botillo del Bierzo’ has an average weight of 1.218 kg and a moisture content slightly above 57% (Table 2). Its protein content is 41.53% DM. These values of moisture and protein align with those obtained by Lorenzo et al. [36] for botillo produced in Galicia (Northwest Spain) and other Spanish sausages like chorizo [37] and salchichón [38]. However, ‘Botillo del Bierzo’ has a fat content of 44.84% of a dry matter basis, higher than that of Galician Botillo, which has an average value of 34%, indicating differences in composition across regions. The PGI ‘Botillo del Bierzo’ specifications require a moisture content below 65%, fat content below 48%, and protein content above 37%, criteria that all producers of ‘Botillo del Bierzo’ comply with [7]. This fat content in ‘Botillo del Bierzo’ is also similar to that of another traditional sausage from northwest Spain called ‘Androlla’, which has fat values ranging from 30 to 46% [36].

The fat content is important, as some authors have recommended reducing the fat levels in Portuguese dry fermented sausages or using collagen casings to lower PAH intake [39,40]. Additionally, Pöhlmann et al. [41] linked PAH levels to the fat content in smoked Frankfurter-type sausages, likely due to the liposoluble nature of these compounds.

Figure 3 presents the results from botillos collected from different processing companies, comparing traditionally smoked samples (24 h of smoking) with unsmoked samples from the same companies. The smoked samples display a slightly higher content of three of the four analysed PAHs than the unsmoked ones, which was expected, although no significant differences were found between the smoked and unsmoked samples. Among the four PAHs, chrysene reached the highest level, while benzo[a]pyrene showed the lowest of the four compounds. These values are slightly higher than those published by Lorenzo et al. [4] for botillo produced in Galicia (NW Spain). Notably, benzo[a]pyrene remained well below the maximum allowable limit of 5 µg/kg for national trade and 2 µg/kg for the EU market. Similarly, the total concentration of the four PAHs was well below the 30 µg/kg threshold set for the national market where it is consumed.

Smoking has been shown to increase PAH levels in meat products, with authors such as Waszkiewicz-Robak et al. [42] quantifying this increase between 22% and 40%. Ciecierska and Obiedzinski [43] further noted that traditional smoking methods contribute to higher PAH contamination levels. In our study, although smoking was performed using the traditional method of direct smoking, the exposure time was brief (only 24 h), resulting in relatively low PAH levels. This contrasts with traditionally smoked sausages in countries like Serbia [13] and Bosnia and Herzegovina [44], where smoking durations are much longer. In some cases, these extended periods cause PAH levels to exceed the limits set by EU regulation No. 835/2011 [19]. However, adjustments to the smoking process allow compliance with the updated EU regulation No. 1327/2014 [20,45], as seen in Poland [46]. These findings confirm what the industrial sector has suspected: that with a short smoking time (24 h), PAH contributions remain minimal. This observation aligns with Skaljac et al. [13], who recommend brief smoking periods to reduce PAH levels while preserving the sensory quality of meat products. Additionally, using collagen casings rather than natural casings can further reduce PAH intake [44]. In the case of ‘Botillo del Bierzo’, it is worth noting that it is typically consumed after several hours of cooking, and the pork caecum casing is discarded. This minimises PAH ingestion, as several studies have indicated that most PAHs produced during smoking remain on the outer casing, with only a small percentage penetrating the interior of the product [39,47].

The analysis of PAH4 in ‘Botillo del Bierzo’ showed significant differences among the producing companies (Table 3). Companies 1, 3, 4, 5, and 6 had the highest PAH levels, with Company No. 1 having the highest concentration (22.46 μg/kg). By contrast, Companies 7, 8, and 10 had significantly lower values (2.49 μg/kg, 3.01 μg/kg, and 1.80 μg/kg, respectively). Of the PAHs analysed, chrysene consistently showed the highest concentration, while benzo[a]pyrene had the lowest in all cases.

Since the smoking time is consistent across all companies and the distances to the smoke emission source are also quite similar (Figure 1 and Figure 4), the differences in PAH levels may be attributed to the use of different types of paprika, particularly concerning their production processes [18]. It is important to note that all botillo samples are safe for consumption, as the permitted limits have not been exceeded in any case (BaP < 5 µg/kg; PAH4 < 30 µg/kg) for the national market, with values for benzo[a]pyrene well below this limit (all values below 1.99 µg/kg), while PAH4 values ranged from 1.80 to 22.46 µg/kg.

The analysis results of the paprika used indicate significant differences among the producing companies. Companies 4 and 5 had the highest PAH4 values (802.32 μg/kg and 788.61 μg/kg, respectively), while Company 2 had a value of 51.45 μg/kg. As mentioned in Section 1, since 2016, dried culinary herbs and spices sold in the EU market have a limit of 10 μg/kg for B(a)P or 50 μg/kg for PAH4. However, smoked fruits of *Capsicum* species are exempt from these limits due to their low consumption rates [28]. Consequently, there are no restrictions on PAH levels in the paprika used for botillo production.

It was observed that companies with higher PAH4 values in their botillo (Companies 1, 4, 6, 5, and 3) also have elevated PAH4 levels in the paprika used (Companies 4 and 5). This correlation can be explained by the paprika’s manufacturing process, which, when added to the botillo during production, contributes varying amounts of PAHs [18]. Furthermore, significant differences in PAH4 levels in botillo were found between the analysed years. The year 2022 recorded the highest PAH4 value (15.81 μg/kg), while 2019 had a lower value (8.69 μg/kg). Similar trends were observed for BaA and BbF, with 2022 exhibiting the highest values and 2019 the lowest (Table 3).

The analysis of the paprika used in the production of ‘Botillo del Bierzo’ was conducted from 2021 to 2024, following the observation of the minimal influence of smoking on PAH levels in ‘Botillo del Bierzo’. Significant differences were also found between the years regarding the paprika. The year 2022 showed the highest values for both botillos and paprikas. Since then, there has been a reduction in PAH4 levels present in the paprika, decreasing from 797.03 μg/kg in 2022 to 378.38 μg/kg by 2024. This reduction is attributed to the efforts made by ‘Botillo del Bierzo’ producers to source paprikas with a lower PAH content while maintaining the desired quality in the botillos.

Regarding the years analysed, a downward trend was observed starting from 2022, when the highest PAH4 value (787.03 μg/kg) was recorded for the paprika used in the production of ‘Botillo del Bierzo’, with the lowest value recorded in 2024 (378.38 μg/kg). As mentioned earlier, this trend can be explained by botillo producers’ efforts to reduce the PAH content by sourcing paprikas with lower smoke content. Velázquez et al. [18] achieved a reduction in PAH4 levels using a dryer system that combusts firewood in a closed chamber with forced convection while maintaining the quality of the smoked paprika.

Table 4 presents the results of PAH4 content according to the type of paprika processing. When analysing the PAH4 content according to the type of paprika, significant differences were observed. Smoked paprika has a higher PAH4 content (811.22 μg/kg) compared with mixed (208.00 μg/kg) and unsmoked paprika (32.99 μg/kg), which is logical since PAHs are introduced to the paprika during the smoking process. Lower values were found for low-smoked paprika (458.51 μg/kg) compared with smoked paprika (811.22 μg/kg), although the difference was not statistically significant. The values obtained for BaP showed a similar pattern, with results for smoked paprika at 83.17 μg/kg versus only 17.63 μg/kg for unsmoked paprika. The values for smoked paprika were slightly lower than those reported by authors, such as Fasano et al. [48], who found BaP values of 104 μg/kg and PAH4 values of 1600 μg/kg for ‘Pimentón de la Vera’. Monago-Maraña et al. [49] reported BaP and PAH4 values of 67 μg/kg and 1780 μg/kg, respectively, in 21 samples of smoked paprika.

The values obtained for unsmoked paprika, with average concentrations of BaP at 17.63 μg/kg and PAH4 at 32.99 μg/kg, align with those reported by Ishizaki et al. [50] for dried red peppers, which had BaP values of 4.5 μg/kg and PAH4 values of 35.5 μg/kg. These values are higher than those reported by Rozentale et al. [51], which ranged from 0.33 to 2.21 μg/kg for BaP and 2.86 to 14.0 μg/kg for PAH4 in samples of paprika and chilli from Brazil and China, along with those obtained by Hwang et al. [52] for paprikas dehydrated by air-drying and heat pump-assisted drying.

It is important to note that smoking is significant in paprika production, as it greatly influences its colouring power [53,54], colour stability [23], antimicrobial and antioxidant properties [55,56], and anti-rancidity effects in dry-fermented sausages [53,57]. However, Coleto et al. [58] showed that the average annual consumption of smoked paprika per person in Spain is 139 g, contributing minimally to PAH intake in the human diet. Similarly, in Romania, Berki et al. [10] reported daily values of 0.02–0.03 g/kg body weight, equivalent to 584–876 g per person per year, which is significantly higher than the average consumption in Spain reported by Coleto et al. [58] but does not pose a serious health concern. Nevertheless, they recommend liquid smoke as a good option to maintain the organoleptic characteristics of smoked paprika while reducing PAH levels.

## 4. Conclusions

‘Botillo del Bierzo’ is a sausage that, due to its composition, is initially more susceptible to accumulating higher PAH levels. However, the 24 h smoking process for ‘Botillo del Bierzo’ contributes minimally to PAH levels in the product. In fact, no differences were found between smoked and unsmoked botillos in the PAHs analysed.

Using paprika with lower PAH content (unsmoked or mixed) enables the production of ‘Botillo del Bierzo’ that adheres to PGI specifications while maintaining the desired organoleptic characteristics provided by the paprika. Producers are also working to reduce PAH content, and in recent years, they have been using paprikas with lower smoking levels for this purpose. According to the 2008 EFSA recommendations, the PAH levels detected in ‘Botillo del Bierzo’ produced under PGI ‘Botillo del Bierzo’ standards do not pose any significant risk to consumers.

## Figures and Tables

**Figure 1 foods-13-04089-f001:**
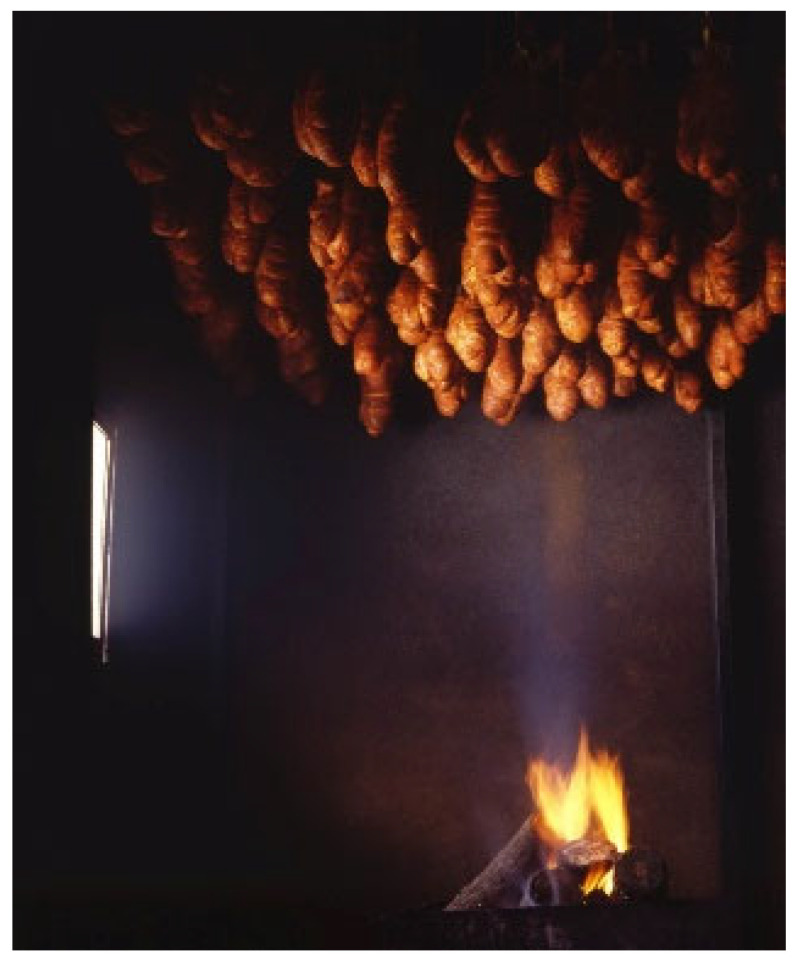
‘Botillo del Bierzo’ during the traditional smoking process, with direct exposure to smoke.

**Figure 2 foods-13-04089-f002:**
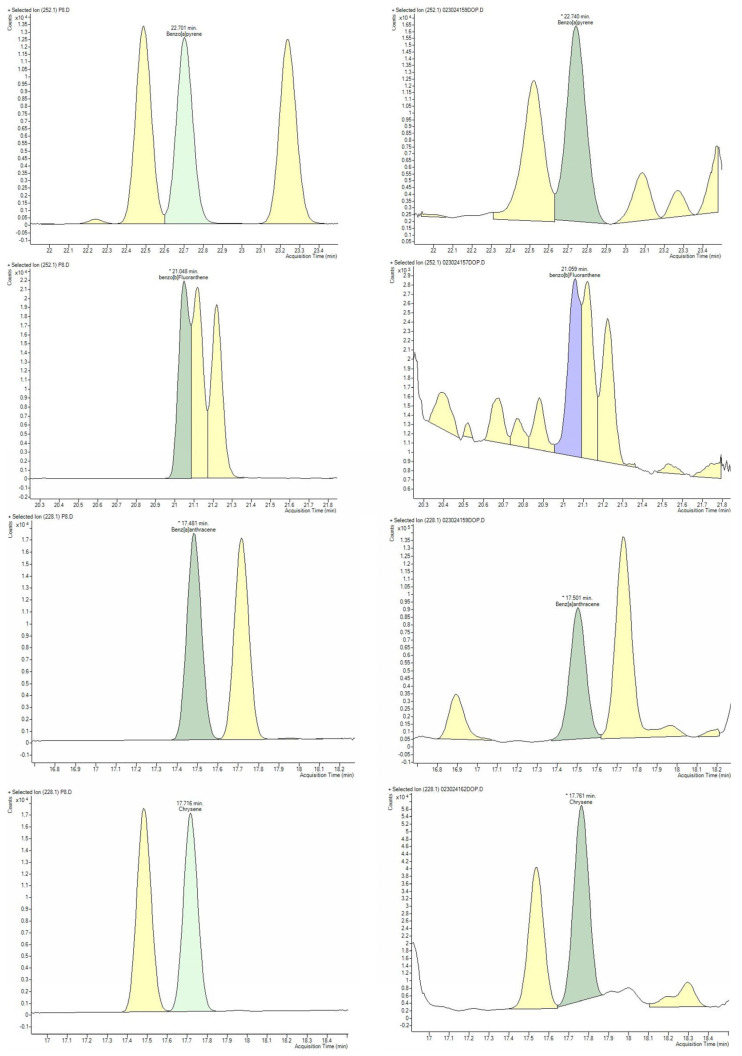
Chromatograms for the standards (**left**) and selected extract of food sample (**right**). The concentration of the standards was 32 µg/kg.

**Figure 3 foods-13-04089-f003:**
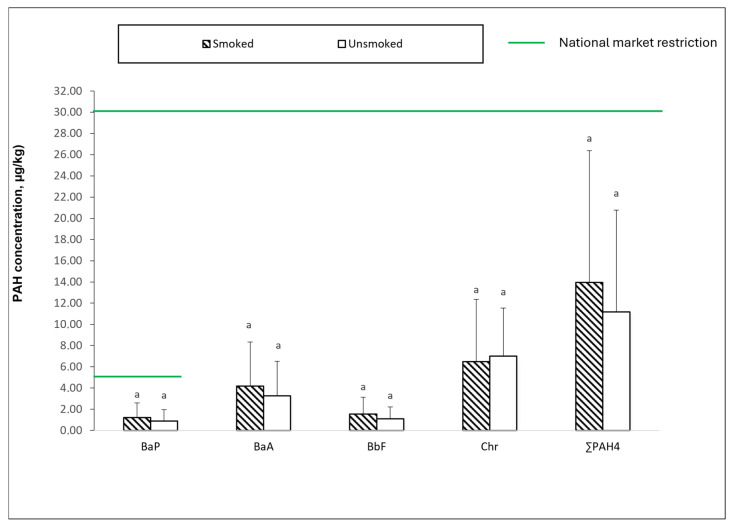
One-way ANOVA results showing the variation in individual concentrations of benzo[a]pyrene (BaP), benz[a]anthracene (BaA), benzo[b]fluoranthene (BbF), and chrysene (Chr) and their combined concentration (∑PAH4) (μg/kg dry weight) in ‘Botillo del Bierzo’ for smoked (1 day) and unsmoked samples. Error bars represent standard deviations of the mean. Identical letters indicate no significant differences between smoked and unsmoked samples (post hoc Tukey test *p* < 0.05).

**Figure 4 foods-13-04089-f004:**
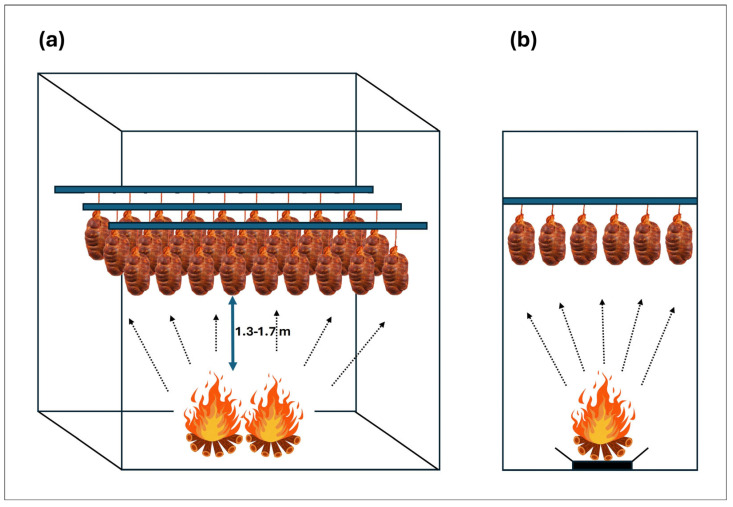
Graphical representation of the direct smoking room: (**a**) 3D view, (**b**) side view.

**Table 1 foods-13-04089-t001:** Validation parameters and results for the GC–MS method used in PAH analysis.

	Benz[a]anthracene	Chrysene	Benzo[b]fluoranthene	Benzo[a]pyrene
Average % RSD_Reproducibility_	10.2	9.8	11.5	9.6
Recovery % Range	75–104	83–97	85–103	65–104
LOQ µg/kg	0.9	0.9	0.9	0.9
LOD µg/kg	0.31	0.32	0.38	0.27
% Uncertainty	20.6	20	22.8	19.4

**Table 2 foods-13-04089-t002:** Characterization of ‘Botillo del Bierzo’ (n = 30).

Weight (kg)	Moisture (g/100 g)	Dry Matter (g/100 g)	Protein (%DM) ^1^	Fat Content (%DM)
1.22 ± 0.22 ^2^	57.39 ± 5.63	42.61 ± 5.63	41.53 ± 3.37	44.84 ± 5.19

^1^ DM: Dry matter. ^2^ Average ± standard deviation.

**Table 3 foods-13-04089-t003:** Two-way ANOVA results for HAPs (μg/kg dry weight) by company and year in ‘Botillo del Bierzo’ and paprika samples (n = 104 and n = 96, respectively), showing the mean concentrations of PAH compounds across different companies and years.

		Botillo						Paprika			
Company	BaP	BaA	BbF	Chr	∑PAHs4	Company	BaP	BaA	BbF	Chr	∑PHA4
1	1.99 a * ± 1.85	7.23 a ± 6.94	2.20 a ± 2.14	11.05 a ± 7.31	22.46 a ± 18.05	1	62.86 a ± 39.44	191.27 ab ± 85.21	68.66 a ± 44.60	291.55 ab ± 111.44	614.33 ab ± 275.65
2	NDab	0.90 bc ± 1.21	NDab	3.00 bcd ± 1.70	4.43 bcd ± 3.77	2	13.00 a ± 12.02	12.15 b ± 1.20	5.65 a ± 4.03	20.65 b ± 5.16	51.45 b ± 9.69
3	1.03 ab ± 0.94	4.67 abc ± 1.51	1.66 ab ± 1.27	7.86 abcd ± 2.26	15.23 ab ± 4.45	3	74.57 a ± 48.35	193.27 ab ± 63.76	65.94 a ± 31.57	306.81 ab ± 111.88	640.59 ab ± 206.31
4	1.45 ab ± 1.41	5.50 ab ± 3.58	1.87 ab ± 1.34	9.16 ab ± 6.02	17.98 ab ± 11.50	4	72.97 a ± 54.59	253.25 a ± 210.61	75.94 a ± 68.97	400.15 a ± 329.24	802.32 a ± 648.72
5	1.12 ab ± 1.29	4.93 abc ± 3.16	1.63 ab ± 1.30	8.01 abcd ± 4.27	15.69 ab ± 8.62	5	99.45 a ± 124.82	250.10 a ± 205.92	84.09 a ± 82.73	354.97 a ± 256.26	788.61 a ± 665.44
6	1.61 ab ± 1.55	4.64 abc ± 3.16	1.75 ab ± 1.46	8.42 abc ± 5.86	16.43 ab ± 9.53	6	64.09 a ± 46.34	154.43 ab ± 93.09	57.67 a ± 42.69	250.36 ab ± 145.02	526.56 ab ± 324.49
7	ND ab	ND c	ND b	1.83 cd ± 1.34	2.49 cd ± 1.38	7	67.33 a ± 37.37	151.93 ab ± 69.56	56.42 a ± 37.69	241.32 ab ± 117.61	517.00 ab ± 236.46
8	ND b	0.91 bc ± 0.99	ND ab	1.79 cd ± 1.50	3.01 cd ± 2.40	8	51.33 a ± 41.06	79.44 ab ± 93.68	28.67 a ± 29.32	106.91 ab ± 128.68	266.36 ab ± 265.83
9	ND ab	2.31 bc ± 2.29	1.00 ab ± 1.22	3.55 abcd ± 3.19	7.58 bcd ± 7.47						
10	ND b	ND c	ND b	1.35 d ± 0.21	1.80 d ± 0.85						
Years						Years					
2019	ND a	2.57 b ± 2.04	ND b	4.57 b ± 2.90	8.69 b ± 6.70	2021	68.06 a ± 47.47	190.22 ab ± 134.27	66.03 ab ± 46.94	306.49 ab ± 213.11	630.80 ab ± 429.52
2020	0.92 a ± 1.33	3.21 b ± 2.82	1.10 b ± 1.05	6.12 ab ± 4.26	11.34 ab ± 8.95	2022	88.21 a ± 98.94	250.06 a ± 206.91	80.62 a ± 75.55	378.14 a ± 290.43	797.03 a ± 658.93
2021	1.12 a ± 1.38	3.91 ab ± 3.68	1.20 b ± 1.55	8.17 a ± 6.10	14.41 ab ± 12.36	2023	63.65 a ± 32.96	202.17 ab ± 146.77	74.20 a ± 49.46	302.52 ab ± 230.41	642.54 ab ± 437.81
2022	1.23 a ± 1.60	5.68 a ± 5.93	2.12 a ± 1.97	6.83 ab ± 7.38	15.81 a ± 16.32	2024	59.09 a ± 47.11	110.68 b ± 93.81	30.04 b ± 30.88	178.57 b ± 158.26	378.38 b ± 307.86

* Means ± standard deviation with the same letter within the same column and factor are not significantly different (post hoc Tukey test *p* < 0.05). HAPs: benzo[a]pyrene (BaP), benz[a]anthracene (BaA), benzo[b]fluoranthene (BbF), and chrysene (Chr); ∑HAP4: sum of benzo[a]pyrene, benz[a]anthracene, benzo[b]fluoranthene, and chrysene. ND: not detected (values below LOQ).

**Table 4 foods-13-04089-t004:** Two-way ANOVA results for type of paprika processing and year on PAH (μg/kg dry weight) content in paprika samples (n = 96), showing the mean concentrations of PAH compounds across different types of processing, excluding year-specific data.

Paprika					
Type of Paprika Processing	BaP	BaA	BbF	Chr	∑PAH4
Smoked	83.17 a * ± 68.51	253.22 a ± 163.30	83.84 a ± 59.06	390.99 a ± 239.74	811.22 a ± 508.95
Low-smoked	68.01 ab ± 42.73	136.16 ab ± 89.01	47.67 ab ± 35.95	206.67 ab ± 143.79	458.51 ab ± 278.98
Mix **	26.90 ab ± 14.43	64.12 b ± 37.24	17.62 ab ± 16.25	99.36 bc ± 57.27	208.00 bc ± 113.25
Unsmoked	17.63 b ± 15.02	3.92 b ± 3.62	4.72 b ± 3.95	6.71 c ± 5.56	32.99 c ± 18.07

* Means ± standard deviation with the same letter within the same column are not significantly different (post hoc Tukey test *p* < 0.05). PAHs: benzo(a)pyrene (BaP), benz(a)anthracene (BaA), benzo(b)fluoranthene (BbF), and chrysene (Chr); ∑HAP4: sum of benzo(a)pyrene, benz(a)anthracene, benzo(b)fluoranthene, and chrysene. ** Mix: mixed paprika (smoked + unsmoked or low-smoked + unsmoked).

## Data Availability

The data presented in this study are not publicly available due to restrictions related to intellectual property and project confidentiality agreements. However, the data are available from the corresponding author upon reasonable request.

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
