# Peer review of "Influence of Smoking and Paprika Spice on the Content of Polycyclic Aromatic Hydrocarbons (PAHs) in the Traditional Spanish Smoked Sausage ‘Botillo del Bierzo’"

_foods, 2024, doi:10.3390/foods13244089_

Round 1

Reviewer 1 Report

Comments and Suggestions for Authors

Great and interesting paper, I would recommend some changes and additions.

- Add unit in Table 2.

- The values ​​of PAH compounds in meat products from different manufacturers and years are not clearly displayed - Do the displayed values ​​of PAH compounds within a company represent the average values ​​of that company over all 4 years? Also, do the displayed values ​​of PAH compounds within a year represent the average values ​​of all companies over a year?

- Line 345 and 346 - You should refer to Table 2., Chr was the highest in 2021, BaP was the highest in 2022

- The values ​​of PAH compounds are not clearly displayed - same questions as for the data shown in table 2.

- It is necessary to add more information about sample preparation, i.e. which part of meat product was tested, whether the casing was removed before taking samples for testing, etc....

Reviewer 2 Report

Comments and Suggestions for Authors

Comments to the Authors

Undertaking the task of determining the concentrations of PAHs and searching for the causes of their increase in meat products seems to be important, especially for foodstuffs consumed very frequently.  Below you will find the list of remarks which I propose to take into consideration while preparing a revised version of the manuscript. 

Introduction

Line 63-68, as well as in the rest of the manuscript: Please standardize the spelling of PAH names. Currently, they are spelled alternately in lowercase and uppercase letters.

2. Materials and Methods

Line 120-122 The subject of the study was not only sausages, but also paprika. Is there more detailed data available on this additive? Do individual producers subject it to their own smoking? Is it purchased by them? Do different manufacturers use the same sources of paprika? What is the difference between smoked and low-smoked paprika?

From where/how were the paprika samples in which PAHs were determined in this study collected?

If I understood correctly, chrysene d12 was used as an internal standard?  In line 152, 189 and 196, the Authors write about internal standard (the abbreviation ISTD appears not until line 197), although it was not stated what it is.

Line 142: The PAH Analyzer calibration sample kit-vial 2  includes several deuterated PAHs. At this point, it can be added that only Chrysene -d12 was included in the study as an internal standard.

Line 149 and 150: With what precision were the samples weighed? Lines 149 and 150 state that ±  0.10 g (i.e. 100 mg) and in line 150, that ± 0.1 mg. Check this and correct it, please.

Line 152-153: I don't understand how “0.150 ug (of internal standard)/ml in each sample replicate” was obtained? Additionally: Whether this applies to the concentration of IS in the sample used for GCMS analysis? 

Line 157: What does the symbol P5PAHs mean? It is not listed among the reagents. What amount (in ug) of standards was used to enrich 1 g of food sample? Were the samples doped at only 1 concentration level?

Line 163: What does the symbol dZsep mean? It is not listed among the reagents.

Line 174-175: Please provide details on the mass spectrometer used. 

Line 176: How long was the splitless time?

Line 175: “PAL RSI 120 detector” ??

PAL RSI 120 is probably an autosampler, please check and correct it. Please provide details on the manufacture.

Line 183: From the literature cited and the text of manuscript in line 191 -193, I conclude that the PAHs determinations were performed by GC-MS/MS technique in MRM (multiple reaction monitoring) system. Please provide more details on this topic.

Line 185: I think it is enough to give the “retention times”. What does the term “relative positions of the compounds” refer to?

 Line 227 – 239: The passage on method validation in line 227 -236 contains new abbreviations of names (D. M. Calibration, D.M. Validation, I.P.) and many numbers.  What does D.M. stand for ? I propose to present the validation procedure, including the results, which are missing in the article, in the form of a table. From the validation results, only the RSDR in the manuscript was provided (Line 252). However, there are no other validation results that were  determined as indicated in line 222.

3. Results and discussion

The descriptions of Tables 2 and 3 are missing the unit of concentration (ug/kg?) and the number of repetitions of each analysis (n)

 Does Table 2 give the company's 2019-2022 averaged results for Botillo and 2021 -2024 for Paprika? In addition, the data for a given year: do they represent the averages obtained for all companies and for all types of Paprika? It would be worthwhile to complete such information in the description of the table. If I am mistaken, please clarify what the results shown are about.

 Table 2 and Table 3 give the same results for Paprika in 2021-2024. Please decide which to leave out, as not to duplicate information.

 Please, supplement the manuscript with chromatograms for the standards and selected extract of food sample.

Conclusions

The article did not examine the different parts of the sausage (interior, casing), so I propose to remove the conclusion in line 405-406.

References

Reference 32 and 33: Please correct the author's name.

References need to be corrected in many places according to the instruction for authors.

Round 2

Reviewer 2 Report

Comments and Suggestions for Authors

Thank you for the clarification and answers.

I propose to move the information about the determination of PAHs in SIM mode from the line 184 to the  part 2.3.3 (line 196- 204). SIM  is not part of a mass spectrometer, but was chosen as the method of determination.

Please re-check that the LOQ values (in ug/kg) in Table 1 are correct, as many of the PAH determination results in Table 3 are below this limit.

Also, please check that in Figure 2 the chromatograms for the standards are not from the right side? Unless chromatograms for low concentrations were selected? It would be useful to state under the figure what concentration of PAHs the selected chromatograms of standards correspond to.

Author Response

Thank you for the clarification and answers.

We sincerely thank you for your thorough review, insightful comments, and valuable clarifications. Your feedback has been instrumental in improving the quality and clarity of our manuscript

I propose to move the information about the determination of PAHs in SIM mode from the line 184 to the  part 2.3.3 (line 196- 204). SIM  is not part of a mass spectrometer, but was chosen as the method of determination.

The requested information has been relocated to the recommended section in the manuscript.

Please re-check that the LOQ values (in ug/kg) in Table 1 are correct, as many of the PAH determination results in Table 3 are below this limit.

The LQ values presented in Table 1 are correct. We have carefully reviewed Table 3 and now the results are consistent with the LQ values. Any values below the LQ were marked as 'ND' (Not Detected).

Also, please check that in Figure 2 the chromatograms for the standards are not from the right side? Unless chromatograms for low concentrations were selected? It would be useful to state under the figure what concentration of PAHs the selected chromatograms of standards correspond to.

It was indeed an error; the chromatograms for the standards were placed on the right side. Figure 2 has been updated accordingly. Chromatograms for the standards are now shown on the left, and those for the selected extract of the food sample are on the right. Additionally, information regarding the concentration of PAHs in the selected chromatograms of the standards has been included. The concentration of the P8 standards shown in the chromatograms is 32 µg/kg.